# Targeting *Aedes aegypti* Metabolism with Next-Generation Insecticides

**DOI:** 10.3390/v15020469

**Published:** 2023-02-08

**Authors:** Michael J. Conway, Douglas P. Haslitt, Benjamin M. Swarts

**Affiliations:** 1Foundational Sciences, Central Michigan University College of Medicine, Mount Pleasant, MI 48859, USA; 2Department of Chemistry and Biochemistry, Central Michigan University, Mount Pleasant, MI 48859, USA; 3Biochemistry, Cell, and Molecular Biology Graduate Programs, Central Michigan University, Mount Pleasant, MI 48859, USA

**Keywords:** *Aedes aegypti*, metabolism, insecticide, dengue, zika, trehalose, trehalase, ecdysone, cholesterol, triacylglycerol

## Abstract

*Aedes aegypti* is the primary vector of dengue virus (DENV), zika virus (ZIKV), and other emerging infectious diseases of concern. A key disease mitigation strategy is vector control, which relies heavily on the use of insecticides. The development of insecticide resistance poses a major threat to public health worldwide. Unfortunately, there is a limited number of chemical compounds available for vector control, and these chemicals can have off-target effects that harm invertebrate and vertebrate species. Fundamental basic science research is needed to identify novel molecular targets that can be exploited for vector control. Next-generation insecticides will have unique mechanisms of action that can be used in combination to limit selection of insecticide resistance. Further, molecular targets will be species-specific and limit off-target effects. Studies have shown that mosquitoes rely on key nutrients during multiple life cycle stages. Targeting metabolic pathways is a promising direction that can deprive mosquitoes of nutrition and interfere with development. Metabolic pathways are also important for the virus life cycle. Here, we review studies that reveal the importance of dietary and stored nutrients during mosquito development and infection and suggest strategies to identify next-generation insecticides with a focus on trehalase inhibitors.

## 1. Introduction

### 1.1. Vector Control Strategies

Vector control strategies can be categorized into chemical, biological, genetic, and environmental methods. Chemical methods rely on general neurotoxins, the four major categories being organochlorines, organophosphate, carbamates, and pyrethroids. Pyrethroids are the current preferred chemical strategy in mosquitoes due to their rapid onset and minimal toxicity in mammalian species [1]. Despite their widespread use, insecticide resistance is a major concern, in addition to off-target effects that can harm invertebrate and vertebrate species, the possibility of occupational exposure, and the unknown impact of sublethal concentrations on vector-borne diseases [1,2,3,4,5,6,7]. 

The main biological method that is being studied is infection of mosquito vectors with the bacterial symbiont *Wolbachia*. *Wolbachia* infection suppresses viral replication with a negligible risk to humans, the environment, and mosquito fitness [8,9,10,11,12,13,14,15,16,17,18,19,20,21]. Logistic and regulatory constraints currently limit the usefulness of this technique [22,23,24,25]. Additional biological methods being studied include leveraging insect viruses and other microorganisms to compete with human pathogens [26,27,28,29,30]. Biological byproducts such as toxins produced by *Bacillus thuringiensis* subspecies *israelensis* (Bti) have been used successfully to control mosquito larvae [31,32]. 

The oldest genetic method is the sterile insect technique (SIT). This method has a long history and relies on radiation-induced chromosome breakage to produce sterile males. The method requires the production of large numbers of sterilized males, which are released into the environment, and mate with a local population of mosquitoes. If the ratio of sterilized males to females is high enough, they will outcompete the non-sterile males and lead to a reduction in the mosquito population [33,34,35,36]. The production of sterilized males is challenging and makes it difficult to translate into largescale mosquito control programs. Oxitec expanded on this concept and used gene editing techniques to produce male mosquitos that carry a mutation that produces infertility in female progeny. This strategy is an improvement over the original SIT because male progeny continues to carry the mutation and pass it onto female progeny [37,38,39,40]. This reduces the number of genetically modified mosquitoes that need to be released into the environment to suppress a mosquito population. 

In addition to SIT and Oxitec techniques, work with transcription activator-like effector nucleases (TALENs) utilized site-specific nucleases to knock out genes with essential functions such as kynurenine 3-monooxygenase (KMO), which produces eye pigmentation in *Aedes* embryos [41,42]. However, because they rely on sensitive DNA-binding interactions, they can be difficult to produce, and scaling up the technology is challenging. The CRISPR/Cas9 approach originally saw issues like TALEN’s with variable site-specific dependence affecting the techniques’ ability for lethality and mutagenesis; however, germline expression of Cas9 overcame the site variability. The inert site serves as a “gene drive” (GD) relying on the presence of an engineered sgRNA with a cas9 exonuclease to allow for precise and rapid dissemination of genetic payloads into mosquito populations [39,43,44,45]. Genetic methods to suppress mosquito populations have tremendous promise, although concerns regarding cost, control, species specificity, and mistrust of scientific and political institutions remain [46,47,48,49,50]. 

Environmental methods to control mosquito vectors are generally safe and sustainable, although they do require community participation and investment. These methods include designing communities to limit accumulation of standing water, and if needed, routine cleaning or covering of water containers. Physical barriers such as bed nets and screened windows can also reduce exposure to mosquito bites. It is also possible to trap mosquitoes and to physically remove them from the environment. 

### 1.2. Insecticide Resistance 

*Aedes aegypti* and *Aedes albopictus* are the primary and secondary vectors for several important arboviruses, including dengue virus (DENV), yellow fever virus (YFV), zika virus (ZIKV), and chikungunya virus (CHIKV) [51,52]. Insecticide application is a critical tool to reduce the population of disease vectors during an outbreak, particularly in the absence of targeted therapeutics and prophylactic vaccines [7,53,54]. Unfortunately, improper use and overuse of insecticides can select mutations that afford resistance to insecticides [7].

Insecticide resistance can arise from different mechanisms, including modification of the target site (i.e., target site resistance). Pyrethroid insecticides function by targeting voltage-gated sodium channels (VGSCs) in the insect’s nervous system. This leads to rapid paralysis and death, often described as “knockdown”. The most well-studied resistance mutations are found in the VGSC gene, and these are described as knockdown resistance (kdr) mutations [1,4]. Kdr mutations are screened in populations so that we can manage the emergence of insecticide resistance. A recent meta-analysis revealed that the rates of the major kdr mutations (i.e., F1534C, V1016G, and S989G) in Asia from 2000 to 2021 were 29%, 26%, and 22%, respectfully. Resistance to dichlorodiphenyltrichloroethane (DDT) was high in both *Ae. aegypti* (68%) and *Ae. albopictus* (64%). Resistance to permethrin (58%) and deltamethrin (27%) was also high in *Ae. aegypti* [55]. A very recent study showed that the L982W mutation, which confers resistance to pyrethroids, was detected at a frequency of >78% in Vietnam and Cambodia. Alleles with concomitant mutations (i.e., L982W, F1534C, V1016G, and F1534C) were confirmed in both countries at a frequency of >90% in Phnom Penh, Cambodia [56]. The spread of insecticide resistance threatens what was once an effective vector control strategy. 

### 1.3. Novel Insecticide Targets

The risk of widespread insecticide resistance requires careful consideration of how these chemicals are used. Integrated Pest Management (IPM) programs leverage information about a pest’s life cycle and their interaction with the environment to minimize the financial costs and impact on human health and the environment. IPM programs promote judicious use of pesticides, although widespread agricultural use of the same pesticides continues to contribute to insecticide resistance [54]. An important area of research is the identification of novel insecticide targets. These studies address the rise of insecticide resistance and offer solutions that can complement current strategies and perhaps develop next-generation insecticides that have minimal off-target effects. 

Identifying insecticide activity in natural products has been a strong focus of recent research, including characterization of components in essential oils, seaweed, botanicals, and medicinal plants [57,58,59,60,61]. Studies have also shown efficacy targeting specific tissues and metabolic pathways, with a focus on new mechanisms of action and avoiding off-target effects. VUO41 and VU730 are inhibitors of mosquito inward rectifier potassium (Kir) channels expressed in the Malpighian tubules of *Anopheles* and *Aedes* mosquitoes. These chemicals are specific to mosquitoes with limited activity against Kir1 channels in mammalian orthologs and honeybees [62,63]. 

Mosquito metabolism and developmental pathways are emerging targets for insecticide development and provide the opportunity to identify insecticides with new mechanisms of action that target molecules specific to a vector of concern (Figure 1). For example, it may be possible to generate next-generation insecticides that target insulin-like peptide (ILP) homologues that promote insulin receptor signaling and drive growth [64,65]. Further, ecdysone receptor (EcR) agonists have shown promise as novel insecticides [66]. Importantly, our study showing that the trehalase inhibitor validamycin A (ValA) inhibits mosquito development and flight supports additional research into metabolic and developmental pathways that can be targeted with next-generation insecticides [67]. 

### 1.4. Summary

Recent research has been expanding the repertoire of pharmacological targets, and basic science research has identified important roles for dietary and stored nutrients in mosquito behavior and development. Importantly, vector-borne diseases rely on a host’s metabolism to propagate, which creates an opportunity to “kill two birds with one stone”. Identification of species-specific targets involved in critical developmental stages will lead to the development of next-generation insecticides and targets for gene drive technology that have new mechanisms of action and that have minimal to non-existent off-target effects. The following is a review of promising research findings related to dietary and stored nutrients and how they are utilized by mosquitoes to drive behavior and development. Further, we discuss how these same nutrients impact infection with important vector-borne diseases and focus on the development of trehalase inhibitors as a future direction. 

## 2. Dietary Nutrients and Impact on Mosquito Behavior and Development 

Dietary nutrients drive mosquito behavior and development; however, the molecular mechanisms involved are largely unknown, which creates an opportunity to identify novel targets for insecticide development and gene drive technology. Importantly, changes in nutrition can lead to far-reaching impacts on mosquito survival, feeding behavior, oviposition, and tolerance to adverse conditions, including exposure to insecticides. Elucidating the metabolic and signaling pathways that control mosquito behavior and development is an important future direction. 

### 2.1. Survival

Dietary nutrients are clearly important for the survival of mosquito larvae and adults. Mosquito larvae depend on organic detritus and adults depend on either plant nectar or vertebrate blood. These diets provide sugar and free fatty acids that can be converted to ATP, protein, and amino acids that are used to make more protein and deoxynucleotides, as well as lipids, which are converted into major insect hormones. Inhibiting any of these major metabolic pathways will negatively impact survival, although identifying a mosquito-specific target will likely be a challenge due to the evolutionary conservation of major biochemical pathways. 

Previous research revealed that mosquitoes can survive on blood-free alternatives, but these contain different mixtures of peptides, amino acids, vitamins, carbohydrates, ATP, bovine serum albumin (BSA), and cholesterol [68,69,70,71]. Partial dietary restriction significantly influences *Ae. aegypti* development. Similar to studies in other biological systems, dietary restriction in both *Ae. aegypti* larvae and adults led to longer lifespans [72]. Specifically, females that were fed only a single or no blood meal survived 30–40% longer than those given weekly blood meals and increasing the concentration of protein in an artificial blood meal led to a decrease in survival. Larvae also lived longer when fed 50% and 25% larval diet [72]. However, this longer lifespan may be due to a developmental arrest, wherein the restriction of larval diet prolonged eclosure time and reduced the size of adult mosquitoes. Larger adult mosquitos also survive longer than small mosquitoes [73]. These data suggest that energy reserves promote development and survival of mosquitoes. In support of this, at low concentrations of larval diet, larvae spend more time foraging for food [74]. A certain threshold of nutrition and energy reserves is likely needed for larvae to progress to the next developmental stage. A recent in vitro study showed that dietary cholesterol mobilized stored triacylglycerol from lipid droplets, which supports that signals received from a mosquito’s diet can promote the utilization of stored nutrients [75].

### 2.2. Egg Production and Oviposition Site Selection

The size of a female mosquito positively correlates with larval diet concentration and nutrient availability and negatively correlates with larval density [73,76,77]. Larvae that have access to high concentrations of dietary nutrients have more energy reserves and can build a larger female adult mosquito. Building a larger female mosquito is important because heavier females exhibit greater blood-feeding capacity, and macronutrients in blood are critical for egg production [73]. Some studies suggest that larger mosquitoes generated from feeding larvae with higher concentrations of larval diet produce more eggs after a blood meal, although this finding is not consistent [73,77]. Although the size of a mosquito correlates with the nutrition they received as larvae, sugar feeding as adults can also modulate egg production. Importantly, high energy reserves and an empty crop correlated with higher egg production, while lower energy reserves and a full crop full of undigested sugar correlated with lower egg production [78]. These studies suggest that dietary nutrients are important for building large mosquitoes that can produce more eggs, which is likely due to the accumulation of stored nutrients. 

### 2.3. Biting Behavior

Dietary nutrients can also modify biting behavior. It is unclear what drives this behavior, although one study revealed that supplementation of blood with 3 and 10% sugar diet significantly increased biting frequency, and continuous availability of a 5% sugar solution increased the probing response [79,80]. Feeding on specific types of sugar also influenced biting behavior [81]. These data suggest that mosquitoes can sense cues from their host, which can drive behavior, and that dietary nutrition can also impact behavior. In a seminal study, a small-molecule agonist of *Ae. aegypti* neuropeptide Y receptor reduced biting likely by signaling that the mosquito was fully engorged or replete with nutrients [82]. It is likely that odorant and neuropeptide receptors serve to select the best source of nutrition and ensure that sufficient nutrition is acquired prior to committing to oogenesis. 

### 2.4. Tolerance to Adverse Conditions

Mosquitoes encounter many stressors in the laboratory and natural environment, and receiving nutrition improves survival. In one study, *Ae. aegypti* were more likely to survive dehydrating conditions if they recently engorged on blood [83]. *Ae. aegypti* were also more likely to escape during an exito-repellency avoidance assay after exposure to deltamethrin and cypermethrin insecticides if they had fed on blood or sugar [84]. A separate study revealed that hydration alone improved survival in a CDC bottle bioassay, suggesting that mosquitoes adequately hydrated with water, sugar water, or blood are more likely to resist insecticide treatment [84,85]. 

## 3. Dietary Nutrients and Impact on Virus Infection

Dengue virus (DENV), Zika virus (ZIKV), yellow fever virus (YFV), and chikungunya virus (CHIKV) are transmitted by *Ae. aegypti*. All viruses are obligate intracellular parasites and depend on the host cell’s metabolism to produce raw materials and chemical energy to produce progeny virions. Studies in animal models have previously linked dietary nutrition and disease outcome. Recent research revealed the importance of host-derived dietary nutrients during arbovirus acquisition in mosquito vectors.

### 3.1. Protein

One study compared ZIKV infection in *Ae. aegypti* that were fed either infectious whole blood or Dulbecco’s phosphate buffered saline solution containing 250 mg/mL BSA. ZIKV acquisition in midgut tissue was worse 7- and 14-days post-infection (dpi) when mosquitoes were fed the infectious protein solution, although dissemination to peripheral tissues was not affected [86]. These data suggest that blood protein is not a critical factor that promotes ZIKV acquisition, and that other blood nutrients may be important for virus infection. 

### 3.2. Sugar 

Mosquitoes acquire sugar in vertebrate blood and plant nectar—mostly sucrose, glucose, and fructose—which are primarily used for the energy demands found in mosquitoes to promote flight, survival, and reproduction [87]. What is less recognized, but also important, is how sugar impacts the virus acquisition in the mosquito. In one study, a sugar meal just prior to administration of an infected blood meal protected mosquitoes from infection with arboviruses from different families [87]. In contrast, a blood meal supplemented with glucose promoted an increase in mosquito infection compared to blood meal alone [88]. These results require further investigation since they are in conflict. It is important to note that administration of a sugar meal would place nutrients into the crop, whereas glucose supplemented into a blood meal would place nutrients into the midgut. These are dramatically different environments, and dietary sugar likely integrates with mosquito metabolism and immunity in unique ways depending on where it resides.

### 3.3. Lipid

The lipid fraction of vertebrate blood is arguably the most complex and contains free fatty acids (FFAs), triacylglycerol, cholesterol, and cell-associated phospholipids. The contribution of blood meal-derived vertebrate lipids to virus infection is largely unknown [8,11,12,13,14]. Previous research has shown that intracellular lipids are important for DENV replication in both vertebrate and invertebrate cells, and that DENV manipulates its host’s lipidome to facilitate replication [15,16,17,18,19,20,21,22]. The importance of lipids in the DENV life cycle is clear, although it is not known how blood-derived lipids impact infection. In mosquitoes, alterations in cholesterol and lipid trafficking through Wolbachia infection or chemical/genetic manipulation interfered with DENV infection [17,18,20,23]. In contrast, human low-density lipoprotein (LDL) inhibited flavivirus infection in vitro and in vivo [8]. Further research revealed that extracellular vesicles (EVs) in serum restricted DENV fusion in the *Ae. aegypti*-derived (Aag2) cell line but not in mammalian cells [13]. Vertebrate lipids appear to inhibit DENV at an early stage in its life cycle and promote DENV at a later stage in its life cycle. In support of this, DENV reduced protein expression of low-density lipoprotein receptor-related protein 1 (LRP-1) in Aag2 cells, leading to increased intracellular cholesterol levels and enhanced virus replication [24]. These studies reveal a complicated relationship with blood-derived lipids, where some species may inhibit acquisition and others facilitate replication and dissemination. 

## 4. Stored Nutrients and Impact on Mosquito Development

Mosquitoes store nutrients in the form of glycogen, triacylglycerols, and trehalose. Glycogen and triacylglycerols are stored in adipocytes in the fat body [89]. Trehalose is synthesized from two glucose molecules and is the major blood sugar in mosquitoes [90]. Mosquito larvae and adults acquire nutrients from either organic detritus, vertebrate blood, or plant nectar, and store nutrients as needed to fuel key life cycle transitions. Recent research has shown the importance of stored nutrients on mosquito development. The molecular mechanisms that contribute to metabolism of stored nutrients are well-established and largely conserved across species, which creates an opportunity to quickly identify targets for insecticide development and gene drive technology. Optimizing compounds that can specifically target *Ae. aegypti* metabolic pathways is an important future direction. 

### 4.1. Lipid 

Fatty acids stored in lipid droplets within cells in the fat body take the form of triacylglycerols and are mobilized through lipolysis to maintain metabolic activity of cells and tissues, and provide energy needs for long-term flight, oogenesis, and resisting starvation [91,92,93,94]. Insect adipokinetic hormone (AKH), juvenile hormone (JH), and 20-hydroxyexdysone (20E) act as metabolic switches to promote the mobilization of stored lipids [95,96]. AKHs activate fat body lipases, which convert triacylglycerols to free fatty acids and glycerol, and allow insects to generate ATP though beta oxidation [95]. JH and 20E regulate expression of genes involved in triacyclglycerol catabolism and β-oxidation [96]. Preliminary studies have shown that dibenzoylhydrazine compounds can function as ecdysone receptor agonists and have insecticidal activity through the promotion of premature development [97,98]. Much of the research that has described insect lipid metabolism has been performed in model organisms, although the limited research available in *Ae. aegypti* supports that many of these pathways and individual genes are highly conserved and would serve as broad targets for the development of next-generation insecticides or gene drive technology. 

Research focused on *Aedes* spp. reveals that triacylglycerol is an important regulator of the gonadotropic cycle—it increases the strength of eggshells and facilities overwintering in diapausing eggs [94]. It is unclear how stored lipids drive the development of mosquito larvae and pupae, although crowding of larvae increases triacylglycerol storage and limits the size of adult mosquitoes and downstream reproductive fitness [99]. Limiting triacylglycerol also reduces the size of male mosquito accessory organs [100]. These data suggest that targeting lipid metabolism would impact multiple life cycle stages, undermining mosquito fitness at the level of larvae, adults, and eggs. 

### 4.2. Sugar 

Transcriptomic studies have shown that metabolism of stored sugars plays an important role in multiple life cycle stages in insects, including eclosure and oogenesis [101]. Insects, including mosquitoes, store sugar in the form of glycogen and trehalose, which have interconnected metabolic pathways (Figure 2A) [89,102,103,104]. Similar to animals, insect glycogen is a branched polymer of glucose that can be degraded as needed to release glucose to support glycolysis and other activities. Glycogen is synthesized in adipocytes in the fat body from UDP-glucose, which itself is generated mainly from dietary carbohydrate or amino acids [89,104]. When required, glycogen breakdown liberates glucose in the form of glucose-1-phosphate that is isomerized to glucose-6-phosphate, with the latter then being utilized in glycolysis or other pathways [89,104]. Glycogen is also stored in eggs and promotes overwintering. Reduction of glycogen levels using RNAi to knock down glycogen synthase kinase-3 (GSK-3) led to embryonic lethality [105,106,107,108]. Glycogen is clearly an important stored nutrient that supports several stages in the mosquito life cycle. 

The other major storage sugar in insects is trehalose, which is the main sugar found in hemolymph. Trehalose, which is also found in fungi, bacteria, and plants, is a non-mammalian disaccharide consisting of two glucose units that are linked together by a 1,1-α,α-glycosidic bond (Figure 2A) [109]. Trehalose is synthesized in fat body adipocytes through trehalose phosphate synthase (TPS)-catalyzed conversion of UDP-glucose and glucose-6-phosphate to trehalose-6-phosphate, which is then dephosphorylated by trehalose-6-phosphate phosphatase (TPP) to give trehalose [104,110]. Trehalose is then secreted via trehalose-specific transporters (e.g., TRET1) into the hemolymph, where it serves as a circulating form of stored glucose [90]. Trehalose is a multifunctional molecule in mosquitoes, as it provides energy and promotes growth, metamorphosis, stress recovery, chitin synthesis, and flight [90,111,112,113,114,115,116,117,118,119,120]. When needed for these functions, trehalose is hydrolyzed by an extracellular trehalose-specific glycoside hydrolase, or trehalase (TREH), yielding two molecules of glucose that can be imported into cells via glucose transporters (GLUTs) [90]. There are two forms of trehalase in insects, including: (i) trehalase 1 (Tre-1), which is soluble and has been identified in insect hemolymph, midgut goblet cells, and eggs; and (ii) trehalase 2 (Tre-2), which is membrane-bound and has been identified in flight muscle, follicle cells, ovary cells, spermatophore, midgut, and brain tissue [90]. 

Because humans and other vertebrates do not synthesize trehalose nor require it as a nutrient, trehalose metabolism represents an attractive target for the development of safe and specific next-generation insecticides. Trehalases have received significant attention as targets because they are required for the mobilization of trehalose into glucose, which is critical to insect physiology [90,121]. For example, multiple RNAi studies have demonstrated that trehalase deficiency causes abnormal growth and development in various insect species, including brown plant hopper and beet armyworm [122,123]. In addition, various small-molecule trehalase inhibitors, such as validamycin A (Val A, Figure 2B), have been shown to impair development at several stages of the insect life cycle [111,112,116,117,119]. With respect to mosquitoes specifically, the prospects of targeting trehalase are also encouraging. Recently, we demonstrated that Val A delays larval and pupal development of *Ae. aegypti*, and it inhibits the flight of adults, likely due to hypoglycemia [67]. 

The promising results of Val A in impairing *Ae. aegypti* development, coupled with the availability of an array of other trehalase inhibitors, motivates continued development and testing of this class of compounds [90]. To complement other existing trehalose mimetics and pseudosaccharides, we recently introduced a variety of synthetic trehalose analogues, a number of which were shown to inhibit bacterial trehalose utilization, including 5-deoxy-5-thio-D-trehalose (5-ThioTre, Figure 2B) [124]. In contrast to Val A, however, 5-ThioTre had no impact on *Ae. aegypti* development or flight [67]. In light of the differential activities of Val A and 5-ThioTre, future research should broadly investigate structure-activity relationships of trehalase inhibitors toward the identification of optimal insecticides for targeting *Ae. aegypti* with minimal off-target effects. In addition, other components of *Ae. aegypti* trehalose metabolism, such as trehalose synthesis via TPS/TPP or transport via TRET1, can also be explored as potential molecular targets. Given the noted absence of trehalose from humans and other vertebrates, such strategies are expected to exhibit higher specificity and limited off-target effects compared to existing insecticides. 

## 5. Stored Nutrients and Impact on Virus Infection

Very few studies utilize *Ae. aegypti* to directly test the role of stored sugars (i.e., glycogen or trehalose) on virus infection, although other mosquito models reveal that stored nutrients are important modulators of infection. One study shows that limiting the concentration of sucrose available to *Culex* mosquitoes reduced total glycogen and lipid per mosquito, and this correlated with an increased ability to orally transmit virus [125]. Another study showed that overexpression of the cellular energy sensor AMP-activated protein kinase (AMPK) reduced glycogen and trehalose concentrations in *Anopheles stephensi*. This led to reduced permissiveness to infection by *Plasmodium falciparum* and a reduction in egg production [126]. Another study using *Anopheles gambiae* showed that knockdown of trehalose transporter AgTreT1 reduced hemolymph trehalose concentration, which correlated with decreased resistance to low humidity, heat, and reduced the number of *Plasmodium falciparum* oocytes [127]. 

One study that used *Ae. aegypti* as a model organism showed that infection with the nematode *Brugia malayi* reduced glycogen and lipid concentration in mosquitoes and reduced maximum flight speed [128]. A set of in vitro and in vivo studies also confirmed the importance of intracellular lipid trafficking for mosquitoes and DENV and identified thiosemicarbazones as a potential class of insecticides that can inhibit *Ae aegypti* sterol carrier protein 2 (SCP-2) [129,130,131,132]. Although not classically considered a stored nutrient, polyamines are formed in a cycle that requires amino acids: L-methionine and L-ornithine. Inhibiting this pathway is detrimental to arbovirus infection [133]. These data are limiting, although the emerging consensus is that the energy and physical material provided by stored nutrients is important for the replication of pathogens and for the cell to mount an immune response to infection, and that competition for this limiting resource can negatively impact mosquito life history traits. 

## 6. Conclusions

The literature focused on insect metabolism is rich and has described the significant impact nutrient deprivation has on mosquito behavior and development. Conserved genes and pathways exist that can be investigated as targets for next-generation insecticides or gene drive technology. A more interesting direction is the identification of unique targets and strategies to control disease vectors while limiting off-target effects. The most promising research areas related to development of new vector control technologies, which include next-generation insecticides, are focused on ecdysone signaling and trehalose metabolism. Ecdysone receptor agonists and trehalose analogues have been used in several insect models and can clearly influence mosquito behavior and development. Interestingly, some trehalose analogues appear to work in bacteria but not in insects [67,134]. This observation opens the door to optimizing next-generation insecticides with enhanced activities and minimal off-target effects. 

## Figures and Tables

**Figure 1 viruses-15-00469-f001:**
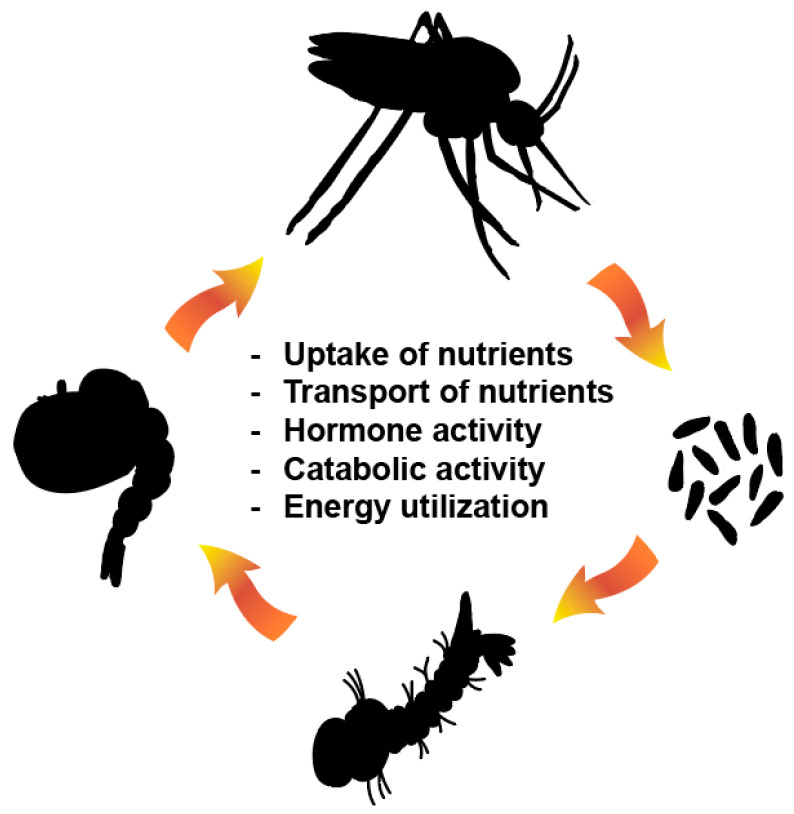
Metabolism of dietary and stored nutrients is critical for each stage in the mosquito life cycle and provides an opportunity to design next-generation insecticides that have minimal off-target effects. Next-generation insecticides can be developed that inhibit uptake of nutrients, transport of nutrients, hormone activity, catabolic activity, and energy utilization pathways.

**Figure 2 viruses-15-00469-f002:**
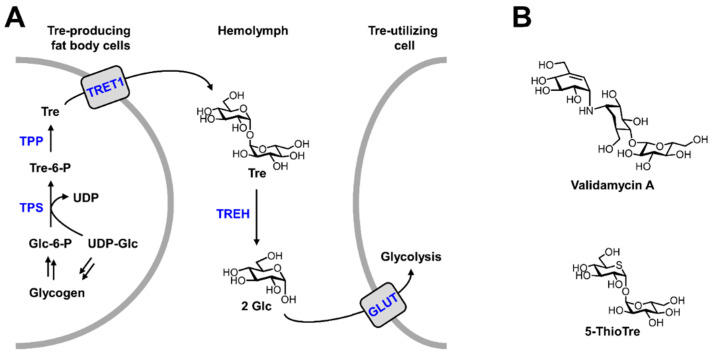
(**A**) Overview of major stored sugar pathways in mosquitoes, with a focus on trehalose metabolism. (**B**) Structures of trehalase inhibitors validamycin A and 5-ThioTre.

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
