# Peer review of "Targeting Aedes aegypti Metabolism with Next-Generation Insecticides"

_viruses, 2023, doi:10.3390/v15020469_

Round 1

Reviewer 1 Report

The review " Targeting Aedes aegypti metabolism with next generation insecticides" provides a concise overview of control strategies used to suppress this species of mosquito and explores the potential of targeting metabolic pathways for alternative insecticidal agents. There are only a few minor comments on the article:

1. The keywords section requires completion.

2. Section 3.2 spelling of arbovirus. Also, this section raises the question of where the virus enters the mosquito. Is it purely through the midgut or could virus enter at any location along the mosquito ailimentary canal?

3. Figure 2, couldn't find the first abbreviation of "trehalase" (TREH).

4. Can the authors clarify "off-target"? Does it mean vertebrate hosts or can it be restricted to non-mosquito or non-dipteran hosts? This might be key to reducing the ecological impact of any agent.

5. "Brugia malayi" in italics.

6. Formatting of references.

Author Response

Response to Reviewers (viruses-2201876)

Reviewer #1:

The review " Targeting Aedes aegypti metabolism with next generation insecticides" provides a concise overview of control strategies used to suppress this species of mosquito and explores the potential of targeting metabolic pathways for alternative insecticidal agents. There are only a few minor comments on the article:

  1. The keywords section requires completion.

This has been completed.

  1. Section 3.2 spelling of arbovirus.

Thank you. We changed arborviruses to arboviruses.

Also, this section raises the question of where the virus enters the mosquito. Is it purely through the midgut or could virus enter at any location along the mosquito ailimentary canal?

The current consensus, which relies on immunocytochemistry and PCR-based techniques suggests that acquisition begins in midgut epithelial cells, and that virus disseminates from this tissue. It is possible that virus that survives digestion moves beyond the midgut to infect other tissues, but this would be secondary to the initial stage of acquisition, and likely occur later since digestion of a blood meal takes 3-4 days. Midgut acquisition is detectable in 18-24 hrs. 

  1. Figure 2, couldn't find the first abbreviation of "trehalase" (TREH).

Thanks. We added, “When needed for these functions, trehalose is hydrolyzed by an extracellular treha-lose-specific glycoside hydrolase, or trehalase (TREH)” near lines 330-331.

  1. Can the authors clarify "off-target"? Does it mean vertebrate hosts or can it be restricted to non-mosquito or non-dipteran hosts? This might be key to reducing the ecological impact of any agent.

In the abstract we say that, “Unfortunately, there is a limited number of chemical compounds available for vector control, and these chemicals can have off-target effects that harm invertebrate and vertebrate species.” This would include honeybees, etc. We added additional clarification in the Introduction, “Despite their widespread use, insecticide resistance is a major concern, in addition to off-target effects that can harm invertebrate and vertebrate species, the possibility of occupational exposure, and the unknown impact of sublethal concentrations on vector-borne diseases.”

  1. "Brugia malayi" in italics.

Thank you! This has been fixed.

  1. Formatting of references.

Thank you. I cleaned up the formatting in this document, although we added new references per the second reviewer and these will need to be incorporated into the formatted list.

Reviewer #2:

The manuscript titled “Targeting Aedes aegypti Metabolism with Next Generation Insecticides” submitted by Conway” reviewed studies that reveal the importance of dietary and stored nutrients during mosquito development and infection and suggested strategies to identify next generation insecticides with a focus on trehalase inhibitors.  For general recommendation, I found the review paper to be overall well written and much of it to be well described. Authors may consiter to (but, not have to) add some information regarding the effects on mosquito development and virus infection of some key metabolites, for examle, spermine (https://doi.org/10.1016/j.chom.2016.06.011) . Actutualy, mosquitoes have the enzyme to metabolize spermine.

Thanks. A short mention of this pathway and its role in virus infection has been included near line 383. Although not classically considered a stored nutrient, polyamines are formed in a cycle that requires amino acids: L-methionine and L-ornithine. Inhibiting this pathway is detrimental to arbovirus infection [132].

Minors:

Page 2, Line 60-62:

“In addition to SIT and Oxitec techniques, work with transcription activator-like effector nucleases (TALENs) utilized site-specific nucleases to knock out genes with essential functions such as eye pigmentation in Aedes embryos [41].”  Authors may provide more specific infor. “In addition to SIT and Oxitec techniques, work with transcription activator-like effector nucleases (TALENs) utilized site-specific nucleases to knock out genes with essential functions such as eye pigmentation in Aedes embryos [41], e.g. the kmo gene (Han, Q., Calvo, E., Marinotti, O., Fang, J., Rizzi, M., James, A.A., Li, J.  Insect molecular biology 2003 v.12 no.5 pp. 483-490)”

Thanks. We added the KMO enzyme and cited this manuscript.

Page 3, Line 123: “ example, it may be possible to generated next generation insecticides that target insulin- …” to “ example, it may be possible to generate next generation insecticides that target insulin-  …”

Thank you. We fixed this typo.

Page 7, Line 295: “Research focused on Aedes spp. reveals ..” here spp. is in normal font

Thanks. We checked all of these in the manuscript and fixed any errors.

Reviewer 2 Report

The manuscript titled “Targeting Aedes aegypti Metabolism with Next Generation Insecticides” submitted by Conway” reviewed studies that reveal the importance of dietary and stored nutrients during mosquito development and infection and suggested strategies to identify next generation insecticides with a focus on trehalase inhibitors.  For general recommendation, I found the review paper to be overall well written and much of it to be well described. Authors may consiter to (but, not have to) add some information regarding the effects on mosquito development and virus infection of some key metabolites, for examle, spermine (https://doi.org/10.1016/j.chom.2016.06.011) . Actutualy, mosquitoes have the enzyme to metabolize spermine.

Minors:

Page 2, Line 60-62:

In addition to SIT and Oxitec techniques, work with transcription activator-like effector nucleases (TALENs) utilized site-specific nucleases to knock out genes with essential functions such as eye pigmentation in Aedes embryos [41].”  Authors may provide more specific infor. In addition to SIT and Oxitec techniques, work with transcription activator-like effector nucleases (TALENs) utilized site-specific nucleases to knock out genes with essential functions such as eye pigmentation in Aedes embryos [41], e.g. the kmo gene (Han, Q., Calvo, E., Marinotti, O., Fang, J., Rizzi, M., James, A.A., Li, J.  Insect molecular biology 2003 v.12 no.5 pp. 483-490)”

Page 3, Line 123: “ example, it may be possible to generated next generation insecticides that target insulin- …” to “ example, it may be possible to generate next generation insecticides that target insulin-  …”

Page 7, Line 295: “Research focused on Aedes spp. reveals ..” here spp. is in normal font

Author Response

(The authors gave the same response as above.)
